

# Natural deep eutectic solvents (NADES) for the extraction of bioactives: emerging opportunities in biorefinery applications

Paula Jauregi[1,2], Leire Esnal-Yeregi[1] and Jalel Labidi[3]

[1] Food Research, AZTI, Basque Research and Technology Alliance (BRTA), Derio, Bizkaia, Spain
[2] Basque Foundation for Science, Ikerbasque, Bilbao, Spain
[3] BioRP Biorefinery Processing Group, Department of Chemical and Environmental Engineering, University of the Basque Country - UPV/EHU, San Sebastian, Gipuzkoa, Spain

## ABSTRACT

Natural deep eutectic solvents (NADES) have emerged as an eco-friendly alternative for extracting bioactives, avoiding the use of flammable organic solvents and extreme temperatures and pH conditions. NADES rely on intermolecular interactions between hydrogen bonding donors (HBD) and hydrogen bonding acceptors (HBA) to form eutectic mixtures with significantly lower melting points than their individual components. These matrices are influenced by factors like water content, temperature, and component ratios. NADES high viscosity can hinder extractive efficiency, which can be mitigated by adding water or working at higher temperatures. However, excessive dilution with water may disrupt the supramolecular structure of NADES, reducing extraction efficiency. A notable feature of NADES is their fine-tunability for specific purposes. Adjusting physicochemical properties such as polarity, pH, and viscosity optimizes extraction efficiency by promoting the solubility of target molecules and interactions between the NADES and target molecules. NADES, unlike organic solvents, can partially disrupt plant and microalgae cell walls, enhancing permeability and extraction efficiency. Moreover, NADES can have a stabilising effect on bioactives and can enhance their biological activity and bioavailability. These attributes, coupled with their low environmental impact in terms of low toxicity and high biodegradability, make NADES attractive for biorefinery applications.

# INTRODUCTION

In the context of increasing concerns for the environmental impact of food and biotechnological processes, the development of sustainable processes are of paramount importance including, circular approaches where, all process streams are reused and/or valorised. Thus, the biorefinery concept is at the center of these developments as it presents an opportunity to obtain high value bioproducts from agri-food by-products and reduce the generation of waste. For example, bioactive compounds, such as phenolics, alkaloids, phenylpropanoids, terpenoids, polysaccharides, lipids, and peptides, offer numerous health benefits, including antioxidant, antimicrobial, antifungal, anti-inflammatory, anti-allergic,

Corresponding author
Paula Jauregi, pjauregi@azti.es

and antitumor effects. These compounds are extracted from natural sources, primarily plant-based biomass, with solvents like hexane and methanol being commonly used for their extraction for application in pharmaceutical and food industries due to their potential health benefits. These conventional organic solvents are frequently employed in chemical processes, but they come with significant drawbacks such as flammability, toxicity, and non-biodegradability, raising environmental concerns. Even ethanol which is considered a green solvent and therefore, can be used in food applications has the disadvantage of high flammability and low selectivity of extraction because of its high solvating power. Natural deep eutectic solvents (NADES) emerge as an alternative ecofriendly extraction method for extraction of a range of bioproducts such as, polyphenols and proteins as it avoids the use of flammable organic solvents, high temperature and/or extreme pH conditions and offers the possibility to modulate the selectivity of extraction by changing NADES components and/or composition. Furthermore, these alternative solvents present an interesting and attractive option compared to traditional organic solvents because of their natural origins, minimal toxicity, and high biodegradability (*Hilali et al., 2024*). Therefore, NADES are gaining much interest for biorefinery applications including, the recovery of bioactives (*Garcia-Roldan, Piriou & Jauregi, 2022*; *Panic et al., 2021*; *Wojeicchowski et al., 2021*). Other interesting applications of DES/NADES include, chemical synthesis (*Cicco et al., 2021*) and biotransformation (*Yang et al., 2017*). Recent reviews on NADES highlight novel applications: (i) health associated applications and in nanotechnology (*Usmani et al., 2023*); (ii) food safety (extraction of pesticides, heavy metals, and others) and biosensors (*Boateng, 2023*); (iii) plant based bioactives extraction (*Zuo et al., 2023*); (iv) plant derived bioactives extraction by a combination of NADES with compressed fluids (*Amador-Luna, Montero & Herrero, 2023*).

This review aims at giving an overview of NADES, their key properties and preparation. Subsequently, particular focus is given to the application of NADES in the extraction of bioactives from sustainable sources such as, food by-products in the context of biorefinery processes. Much attention has been given to this topic in recent years however, there is not much published yet on the extraction of proteins and the extraction of bioactives from microalgae, which is a promising and sustainable source of bioactives. Thus, here an overview of applications of NADES to the extraction of polyphenols and proteins from plant-based by-products, and bioactives from microalgae is presented. Particular emphasis is given to analysing the selectivity of the separation in relation to both, the physicochemical properties of NADES and the bioactives. This will lead to an improve understanding of the selectivity of the separation and mechanism of the extraction which, can help with the design of effective separations. Moreover, other aspects and key properties of NADES are highlighted such as, their stabilizing effect on bioactives and their environmental impact which, should also be taken into account as well as their effectiveness when designing an extraction process.

## SURVEY METHODOLOGY

The research topic was chosen after conducting a thorough review of existing literature and identifying gaps in current knowledge. Emphasis was placed on relevant and emerging areas

within the field of biorefinery, with a specific focus on the valorization of byproducts using NADES solvents. The main databases search were: Science Direct, Google Scholar and Web of Science, and the following keywords and combinations were used for a bibliographic search: "extraction" AND "NADES" AND "bioactives" AND "by-product"; "extraction" AND "microalgae" AND "NADES" AND "protein" OR "pigments"; "NADES" OR "DES" AND "physicochemical properties"; "NADES" AND "extraction" AND "proteins" OR "polyphenols" AND "biorefinery application".

The research period spanned the last ten years up to the present date, although articles outside of this timeframe were cited if they were pivotal references in the field under study. This timeframe was chosen to comprehensively address pertinent studies and observe potential changes or trends over time.

The identified topics were organized thematically to facilitate understanding and analysis. This thematic approach allowed for the grouping of similar studies and effective comparison of results. For example, in the "Extraction of Proteins" section, various studies have been compared regarding protein extraction from by-products, as reported by different authors.

The review process was divided into several stages. Initially, a preliminary literature review was conducted to identify relevant studies. Subsequently, inclusion and exclusion criteria were applied to select the final articles. The criteria followed for the selection of articles was: (1) relevance (2) quality of information and presentation, in that order. Mostly primary sources of information were used, and secondary sources were used only on a few occasions particularly, as an aid to find specific and detailed collected information.

## NADES and their preparation

Deep eutectic solvents (DES) are a blend of multiple compounds that form associations through hydrogen bonding. Typically, DES are created through the complexation of a quaternary ammonium salt, a hydrogen bond acceptor (HBA) with a metal salt or a hydrogen bond donor (HBD). The salt constituents in DES typically possess low lattice energy (*Kovacs et al., 2020*). These components establish a hydrogen bonding network, enhancing the system's stability. In the case of ionic constituents, this network facilitates charge delocalization, and the reduction in melting point is a consequence of these combined effects (*Kovacs et al., 2020*).

NADES are bio-based DES which are derived from natural components such as, aqueous solutions (25–50% water) of choline chloride, sugars, and polyols. Comprising diverse cellular constituents (metabolites) and other naturally available sources, NADES inherit the advantageous characteristics of DES as a subclass (*Fan et al., 2023*). *Choi et al. (2011)* hypothesized that NADES occur naturally within cells due to the abundance of compounds like amino acids, sugars, organic acids, and choline, and this leads to the creation of a unique liquid type alongside water and lipids. Therefore, they could have an important role in the biochemistry of cells and organisms.

NADES are formed from a eutectic mixture of Lewis or Brønsted acids/bases (see Table 1 for some examples). NADES media primarily depends on the intermolecular interactions among their components, a HBD and a HBA, to form a eutectic mixture with a final

**Table 1** Some of the most used NADES of different polarities for the extraction of bioactives.

| | HBA | HBD | Aplication | Reference |
|---|---|---|---|---|
| Hydrophilic | Choline chloride | Glucose | Anthocyanin from blueberry peel | *Grillo et al. (2020)* |
| | | | Phenolic from grape skin | *Radošević et al. (2016)* |
| | | Fructose | Phenolic from grape skin | *Radošević et al. (2016)* |
| | | Xylose | Phenolic from grape skin | *Radošević et al. (2016)* |
| | | Glycerol | Polyphenols from orange peel. | *Ozturk, Parkinson & Gonzalez-Miquel (2018)* |
| | | | Anthocyanin from blueberry peel | *Grillo et al. (2020)* |
| | | | -Phenolic from grape skin. | *Radošević et al. (2016)* |
| | | Ethylene glycol | Polyphenols, proteins and D-limonene from orange peel waste | *Panic et al. (2021)* |
| | | 1–2-Propanediol | Polyphenols and protein from spent coffee ground | *Garcia-Roldan, Piriou & Jauregi (2022)* |
| | | | Phenolic from rosemary leaves | *Wojeicchowski et al. (2021)* |
| | | | Chlorophyll and carotenoids from microalgae | *Cicci, Sed & M (2017)* |
| | | Citric acid | Anthocyanin from blueberry peel | *Grillo et al. (2020)* |
| | | Lactic acid | | |
| | | Malic acid | | |
| | | Levulinic acid | Protein from bamboo shoot | *Lin et al. (2020)* |
| | | Oxalic acid | Protein from cod skins | *Bai, Wei & Ren (2017)* |
| | | Acetic acid | Protein from pomegranate peel | *Hernández-Corroto et al. (2020)* |
| | Betaine | Triethylene glycol | Polyphenols from spent coffee ground | *Garcia-Roldan, Piriou & Jauregi (2022)*. |
| | | Glycerol | Phycocyanin, chlorophyll and fatty acids from microalgae | *Wils et al. (2021)* |
| | Glucose | Glycerol | Protein from orange peel waste | *Panic et al. (2021)* |
| | | | Phycocyanin, carotenoids, and fatty acids from microalgae | *Hilali et al. (2024)* |
| | | Ethylene glycol | Protein from orange peel waste | *Panic et al. (2021)* |
| Hydrophobic | Menthol | 1,2-Octanediol  1,3-Propanediol  Octanoic acid  Lauric acid | Fatty acids, chlorophylls and carotenoids from microalgae | *Hilali et al. (2024)* |

**Figure 1** Hydrogen bonding between NADES constituents.

melting point much lower than the individual components (*Grillo et al., 2021*). This makes NADES matrices readily influenced by various factors, such as water content, temperature, and component ratio (*Liu et al., 2018*). The interaction between these compounds mainly through intermolecular hydrogen bonds, along with other intermolecular interactions like van der Waals and electrostatic forces results in a supramolecular structure (Fig. 1). Water can act as both, HBA and HBD and thus, will interfere with the hydrogen bonding of the NADES components. At water content higher than 50%, NADES may be considered an aqueous solution to its constituents rather than a eutectic mixture (*Dai et al., 2015*). Therefore, water in some proportion can strengthen the hydrogen bonds of the NADES system but in high proportion, it will lead to the disruption of the supramolecular structure, potentially affecting the extraction capacity of NADES. Nonetheless, this is still not well understood.

Other ecofriendly solvents that have gained attention are ionic liquids (IL) with some similarities with DES. IL are formed from the systems composed primarily of one type of discrete anions and cations (*Smith, Abbott & Ryder, 2014*). Their mixture leads to a lowering of their melting point also. However, while IL offer favorable thermodynamic qualities, their potential toxicity and environmental persistence raise concerns. DES share thermodynamic characteristics with IL but have distinct advantages. They are easily synthesized, more cost-effective, environmentally friendlier, and exhibit lower toxicity (*Benvenutti, Zielinski & Ferreira, 2019*). According to some reported estimations, the cost to synthesize DES was only 20% of that of IL and the DES components were ten times less expensive than those of IL (*Grillo et al., 2021*). Also, DES can be recycled more readily than IL.

One of the advantages of DES/NADES is the simplicity of their preparation. Another important advantage in relation to the IL is that in their preparation no by-products or waste is generated thus, no need for further purification. The first DES formed by *Abbott et al. (2003)*, and one of the most studied type was synthesized by mixing a quaternary ammonium salt (HBA) with high melting point such as, choline chloride (ChCl) with an HBD such as, urea which results in the formation of the hydrogen bonds between these two compounds with the subsequent disruption of the crystalline structure resulting in a depression of the melting point. For example, when ChCl (melting point, 302 °C) and urea (melting point, 133 °C) are mixed in a molar ratio of 1:2, a eutectic mixture is formed with the melting point of 12 °C, which is much lower than each component (*Abbott et al., 2003*). Subsequently, it has been shown that quaternary ammonium salts can form eutectic mixtures with various amide compounds with similar solvent properties to IL at room temperature (*Grillo et al., 2021*).

Nevertheless, DES formulations have not consistently met these ecological and toxicological standards (*Hilali et al., 2024*), leading to the interest in NADES as an alternative. Those solvents can be prepared by a combination of a wide range of natural compounds (see Table 1 for some examples). *Panić et al. (2020)* investigated the use of a range of NADES from cheap, readily available, biodegradable, and biocompatible components (ChCl, sugars, and polyalcohols) for biorefinery applications. As a recent innovation *Picchio et al. (2022)* synthesized a series of novel NADES based on ChCl and plant-derived polyphenols for their application as bioadhesives and in corrosion protection. ChCl based NADES are particularly interesting for food applications given its low cost, high biodegradability, and low toxicity (*Boateng, 2023*; *Radosevic et al., 2015*).

The multitude of potential combinations in NADES formulation provides endless opportunities for tailoring extraction techniques to specific needs. Therefore, several studies have suggested using the COSMO-RS model as an alternative to traditional experimental methods for selecting solvents in the preliminary stages of the solvent selection process. For instance, *Wojeicchowski et al. (2021)* demonstrated that COSMO-RS could accurately evaluate the relative solubilities of rosemary biocompounds, which exhibited a strong correlation with total phenolic content values in DES-based extracts (*Wojeicchowski et al., 2021*). Furthermore, in the study by *Jeliński & Cysewski (2018)*, this model predicted the solubility of rutin in NADES, facilitating the identification of the most efficient NADES for solvating rutin.These studies highlight the potential of using the COSMO-RS model as a tool for solvent selection in the extraction of bioproducts from natural sources and emphasize the importance of choosing the right solvent to optimize the extraction process and improve the yield of valuable compounds.

*Benvenutti, Zielinski & Ferreira (2019)*, in their review, presented two methods for creating NADES. In the first method, the compounds are diluted in water, stirred, heated to 50 °C and then evaporated under vacuum. The resulting liquid is dried using silica gel until a constant weight is achieved. The second method involves mixing components, including water, and heating them to obtain a clear liquid. A similar method was described by *Garcia-Roldan, Piriou & Jauregi (2022)* where NADES were prepared by mixing betaine and triethylene glycol at a molar ratio of 1:2 for 30 min at 80 °C. Subsequently, 30% or

40% of deionized water was added, and the mixture was stirred using magnetic agitation until a colorless solution was obtained.

Typically, the sample (food matrix/by-product) is put in contact with NADES and mixed by stirring (*Garcia-Roldan, Piriou & Jauregi, 2022*) or other mechanical aid such as, microwave radiation or sonication. The main operating parameters for a given NADES extraction are: (i) sample to solvent ratio (ii) temperature (iii) time. NADES are typically being used as a solid/liquid extraction method; however, they has been also used as part of an aqueous two-phase system for liquid-liquid extraction of proteins (*Xu et al., 2015*).

## NADES physicochemical properties

An important feature of NADES is the supramolecular structure created by the hydrogen bonding between the HBA and the HBD. In the case of the DES/NADES formed by a quaternary ammonium salt such as, ChCl as HBA and an HBD (*e.g.*: alcohols or sugars) the formation of the hydrogen bonds between these two compounds leads to the delocalization of the charge of the anion (chloride) in the salt and reduction of the ionic forces between this and the cation in the salt (Fig. 1). This disruption of the crystalline structure leads to the formation of a new suprastructure held together mainly by hydrogen bonds which is more loose, less structured and less dense than the former resulting in a depression of the melting point. The number of hydrogen bonds and strength of intermolecular interactions will have an impact on the physicochemical properties of the NADES. Thus, the type and composition of NADES as well as the water composition, will have an effect on these properties and these in turn on the extraction capacity of NADES. In addition, the chemical tunability of these solvents allows them to be finely adjusted to suit specific applications, owing to the extensive options for combining different compounds and adjusting molar ratios to form these solvents. Physicochemical properties of NADES are mainly associated to the HBD component (*Jurić et al., 2021*). In this context, the main physicochemical properties of NADES including, viscosity, polarity, molar ratio, and pH are introduced and discussed below.

### Viscosity (water composition)

High viscosity entails significant limitations to the mass and energy transfer during chemical reactions therefore, limiting the viscosity of NADES is necessary by selecting smaller constituent molecules with fewer hydrogen bond donating/accepting groups and weaker interaction (*Kovacs et al., 2020*). Thus, strong hydrogen bond network formation and interactions leads to high viscosity.

The high viscosity of NADES (typically 200–500 cP at 40 °C) (*Grillo et al., 2021*) can hinder their extractive efficiency which, can be reduced by adding a certain percentage of water or by working at higher temperatures. However, high dilutions with water can result in a disruption of the supramolecular structure of NADES as water molecules will compete with NADES components for hydrogen bonding. The introduction of water creates new hydrogen bonds between the NADES components and water molecules, increasing available space in the solvent mixture (*Ling et al., 2020*). This was also supported by a detailed study by *Alcalde et al. (2019)*.

These structural changes allow for better mass transfer and more effective interaction with the target substances (*Ling & Hadinoto, 2022*). The result is an enhanced solubility, making NADES with adjusted water content valuable for various applications, especially in the extraction of bioactive compounds (*Ling et al., 2020*). However, *Dai et al. (2015)* showed that when the water content in a NADES exceeds 50%, it may transition from being a eutectic mixture to an aqueous solution. This shift in composition results in significant changes in the physicochemical properties of the solution. Additionally, the solubility capacity of the mixture can be affected, and it may behave more like a conventional aqueous solution rather than a NADES. For example, *Ling et al. (2020)* in their study observed that the ChCl:ascorbic acid NADES system with 10% of water exhibits the highest solubility for antioxidant extracts. In contrast, when the mentioned system contains 50% water, the solubility of antioxidant extracts is like that of pure water. Hence, by introducing a specific percentage of water, it is possible to decrease the viscosity of a NADES system, thereby improving the mass transfer during the extraction process. Nonetheless, excessive dilution with water can lead to the disruption of the supramolecular structure within the NADES, consequently having an adverse effect on the solubility of the extract.

Density and viscosity are mostly discussed together, as they show a similar relation to the intermolecular interaction energy and temperature (*Kovacs et al., 2020*). Higher interaction energy increases these properties, while higher temperature results in a decrease of these properties. Thus, increasing interaction between NADES components yields higher viscosity and density. Viscosity has been identified to have a connection with electronic conductivity of NADES. As described by *Dai et al. (2015)*, the higher conductivity of the ChCl-alcohol than the ChCl-sugar NADES could be ascribed to the lower viscosity of the alcohol than the sugar component of the NADES. Thus, lower viscosity leads to higher conductivity as observed also when studied the water effect on NADES (*Dai et al., 2015*). In addition, an increased concentration of charge carrying species (salts) has a positive effect on conductivity (*Kovacs et al., 2020*); this was also demonstrated by *Dai et al. (2015)*. Moreover, an increase in temperature (as it decreases the viscosity) increases the conductivity. The molar ratio of constituents also affects the conductivity through its effect on the viscosity.

*Alcalde et al. (2019)* found a non-linear relationship between temperature and electrical conductivity across various studied molar ratios and water contents.

### Polarity

Polarity is a critical factor influencing the solvation capabilities of solvents. *Ling et al. (2020)* suggested that the solubility of antioxidant extracts is influenced by the polarity of NADES system, where the polarity can be affected by the addition of water. They found that the addition of 10% of water to the DES increased the polarity and further enhanced the solubility of antioxidant extracts. Additionally, the work by *Xu et al. (2019)* particularly illustrates the effect of NADES polarity on extraction efficiency. They investigated the extraction of flavonoids with different polarities from citrus peels with NADES of varying polarities and found that the strong dependence of extraction yield on HBD polarity could be explained by the principle of like dissolves like. This is in agreement with *Panic et*

*al. (2021)* who found little differences in polyphenols yields between the NADES tested as these NADES had similar polarities, and with *Dai et al. (2013)* who analyzed that the NADES with the lowest polarity showed the lowest efficiency for polar compounds, but high efficiency for nonpolar compounds.

### pH

HBD within NADES can influence the pH of the solution and so, have a significant impact on their behavior and performance. For example, *Hou et al. (2018)* applied and compared choline based and lactic acid based DES to delignification of rice straw and they found that most efficient delignification was obtained with ChCl polyols and lactic acid amides DES, which had more basic pH; as lignin is an alkaline soluble biopolymer its extraction will be favored at alkaline pH. Furthermore, *Panic et al. (2021)* observed that polyol-based DES are excellent solvents for extraction of polyphenols considering their lightly acidic pH value (which is considered the optimum pH for polyphenols extraction) and low viscosity. This may be since the acidity or alkalinity of the solution can influence the ionization states of the various components within the solvent and, this can affect the solvent's properties and interactions with other substances. Consequently, under specific pH conditions, there is a propensity for enhanced interactions between the elements of the NADES and the extract. For instance, electrostatic forces may come into play, ultimately amplifying the extraction efficiency.

### Melting point (molar ratio)

In many studies, different ratios of HBA:HBD are used, even though the thermodynamic definition suggests there should be a single mixing point called the eutectic point (*Alcalde et al., 2019*) of a given HBA:HBD ratio. The molar ratio of HBA to HBD is a critical factor influencing the melting point of NADES as it will affect the strength of interaction between them; the higher the strength of interaction the bigger the reduction in the melting point. *Ling & Hadinoto (2022)* demonstrated that varying HBA:HBD ratios in ChCl:urea resulted in significant differences in melting points. A 1:1 ratio produced a NADES with a high melting point (above 50 °C), while a 1:2 ratio yielded a much lower melting point (around 12 °C). Furthermore, the choice of HBD component is equally crucial. Various HBD, including citric acid, malonic acid, oxalic acid, glycerol, ethylene glycol, and xylitol, can lead to DES with melting temperatures ranging from 69 °C to room temperature. Moreover, *Kovacs et al. (2020)* noted that the melting point decreases with the asymmetry of the cation, and the rising electron affinity of the anion (as this will lead to stronger H bonds between the constituents) also contributes to a lower melting point. Hence, both the HBA:HBD molar ratio and the specific HBD component play pivotal roles in tailoring the physical properties of NADES, including the melting point, to meet the requirements of various applications.

Another interesting property of NADES is their cell disintegration property, which is crucial for facilitating the extraction of compounds from plant cells. Their supramolecular structure forms hydrogen bonds with the cell constituents, partially disrupting plant cell walls during these applications (*Yang et al., 2017*). NADESs property of cell disintegration has been shown of high importance for both plant-mediated biotransformation and

extraction of various components from plant cells and tissues (*Yang et al., 2017*). *Ozturk, Parkinson & Gonzalez-Miquel (2018)* investigated the extraction of antioxidants from orange peel using ChCl based NADES and compared with ethanol. They found that NADES solvents were similar or superior to ethanol in extraction efficiency and the corresponding biomass showed a higher disintegration level than that of ethanol, as visualised by scanning electron microscopy. Similarly, *Fan et al. (2023)* found significant changes in the plant powder morphology by scanning electron microscopy after NADES treatment which, suggests NADES mediated cellulose degradation.

In summary, physicochemical properties of NADES can be modified based on the choice of HBA and HBD, their molar ratio and water composition. These in turn, will have an impact on the extraction capacity and selectivity of NADES. Following the principle of like dissolves like the extraction of the bioactives should be optimized by choosing NADES that maximize their solubility. Yet there is not complete understanding of the fundamentals of the extraction efficiency and selectivity of bioactives from food by-products.

## NADES environmental impact

Only a few studies have been reported on the environmental impact of NADES including, sustainability and toxicity assessments. *Zaib et al. (2022)* carried out the environmental impact of reline, a NADES made of ChCl and urea, based on its production and on a life-cycle assessment of its use as chemical reaction medium in comparison with other conventional solvents. They concluded that ethanol and methanol were more environmentally friendly than reline although as they stated, other considerations such as, flammability and toxicity must be considered. In addition, in the case of biorefinery processes other considerations such as, whether the solvent is food grade and other beneficial effects such as, improved bioactivity/bioavailability of the recovered compound must be taken into account. In addition, *Zaib et al. (2022)* compared the environmental performance of reline with other DES made on ChCl but with different HBD components including citric acid. They concluded that ChCl with citric acid had the highest environmental impact due to the citric acid production by fermentation.

### *Cytotoxicity of NADES*

The cytotoxicity of NADES is currently under extensive examination in relation to various human cell lines. Notably, research by *Radošević et al. (2016)* discovered that NADES based on ChCl, containing sugars and organic acids exhibited low cytotoxicity, being ChCl:malic acid NADES system the one that largely influenced cell viability, as showed by *Radosevic et al. (2016)* in another study. Furthermore, *Radosevic et al. (2015)* findings indicated that ChCl in combination with glucose and glycerol showed low cytotoxicity, whereas ChCl combined with oxalic acid displayed moderate cytotoxicity. They suggested that these could be related to the formation of cell damaging calcium-oxalate crystals. Those results were supported by *Hayyan et al. (2016)* as they observed that NADES containing fructose, sucrose, glucose and glycerol had low toxicity, while malic acid–based NADES exhibited the highest toxicity.

*Mohd Fuad, Mohd Nadzir & Harun@Kamaruddin (2021)* suggested that HBD with acidic properties can deactivate cells by causing protein denaturation at the cell wall,

leading to cell collapse and death. Moreover, *Koh et al. (2023)* mentioned in their study that the sugar containing NADES was metabolized to provide energy to the metabolic pathway that helped maintain the integrity of carbohydrates for cell growth. Therefore, studies have shown that NADES containing organic acids are more toxic than those based on sugars and polyols, but this does not mean that all organic acid–based NADES are highly toxic.

Additionally, the toxicity of NADES is believed to be linked to the charge delocalization process resulting from the formation of hydrogen bonds so, a change in the molar ratio is expected to have a direct impact on their toxicity (*Mohd Fuad, Mohd Nadzir & Harun@Kamaruddin, 2021*). In this way, *Ahmadi et al. (2018)* found that the molar ratio played a pivotal role in determining the cytotoxicity of ChCl-based NADES against the HEK-293 cell line, although the same trend was not observed in different NADES. According to *Radosevic et al. (2016)*, the slight difference between cell viability of channel catfish ovary cells treated with ChCl:malic acid 1:1 and those treated with ChCl:malic acid 1.5:1 showed that the cytotoxicity increased with the increase in acid content. This observation highlights the importance of considering the specific molar ratio when assessing the toxicity of NADES.

On the other hand, *Hayyan et al. (2016)* found a correlation between viscosity and toxicity. They found that the less viscous NADES (ChCl:glycerol) exhibited the lowest toxicity, while the most viscous NADES (ChCl:malic acid) showed high toxicity.

### Biodegradability of NADES

There are only a few studies addressing the biodegradation of NADES. The biodegradability of NADES can be tested by a closed-bottle test, according to the Organisation for Economic Co-operation and Development (OECD) test guidelines, whereby it is considered "readily biodegradable" if there is a 60% removal of the theoretical oxygen demand within 10 days over a 28-day period (*Koh et al., 2023*). In this context, *Huang et al. (2017)* assessed the biodegradability of NADES derived from ChCl and glycerol combined with alcohols, sugars, or amino acids. They showed that the biodegradability for all the tested NADESs was >70% after 28 d, so they classified all NADES as "readily biodegradable" according to OECD-criteria. Furthermore, *Radosevic et al. (2015)* showed that biodegradability of ChCl-based DESs with glucose, glycerol and oxalic acid was more than 60% and therefore all of them can be referred to as 'readily biodegradable'. The highest level of biodegradation was observed in ChCl:glycerol (96%) while ChCl:oxalic acid (68%) exhibited the lowest level of biodegradation.

In a separate study, *Wen et al. (2015)* found that only two NADES were 'readily biodegradable'. They showed that ChCl based DES exhibited higher biodegradability compared to those based on choline acetate. Additionally, DES containing urea and acetamide as the HBD were more susceptible to biodegradation in comparison to those containing glycerol and ethylene glycol.

## NADES biorefinery applications

The exploitation of biomass such as, crops, algae, by-products to obtain a wide range of industrially important products is at the center of the implementation of bioeconomy by

replacing fossil-based resources for biomass resources. In this context, the development of biorefinery processes that aim at producing diverse products and/or energy from biomass while maximizing the use and/or valorising all process streams subsequently, reducing/eliminating waste generation are being sought. As an example, see *Rodrigues, Gando-Ferreira & Quina (2022)* for examples of biorefinery processes applied to winery by-products. Moreover, there is an increasing demand for naturally produced additives in the food industry such as, preservatives and antioxidants as well as a demand for food ingredients with additional biological functionalities or bioactivities. Plant based by-products generated after processing such as, grape marc, coffee spent ground, and brewer spent ground are a rich source of a wide range of bioactive compounds such as, phenolics, proteins and peptides (*Iriondo-DeHond, Iriondo-DeHond & Del Castillo, 2020*; *Rodrigues, Gando-Ferreira & Quina, 2022*; *San Martin et al., 2021*). Moreover, microalgae can also be considered a sustainable source of alternative protein and of other compounds of interest such as, pigments and lipids. Typically, these bioactives are obtained from the different process streams by applying solid–liquid extractions using hydroalcoholic solvents. Therefore, developing green solvent extraction methods is of utmost importance to reduce the use of flammable organic solvents and apply milder operating conditions. In this context, NADES emerge as an alternative ecofriendly extraction method for bioactive compounds extraction (*Garcia-Roldan, Piriou & Jauregi, 2022*; *Grillo et al., 2020*; *Ozturk, Parkinson & Gonzalez-Miquel, 2018*; *Panic et al., 2021*).

The key features of NADES that make them attractive as extractants for these applications are their: (i) high tuneability (ii) biodegradability and low toxicity (ii) stabilising effect (iii) effect on enhancing bioactivities and bioavailability (iv) ability to partially disrupt plant cell wall (v) selectivity.

These properties of NADES are further explored and illustrated with examples of their application in biorefinery (see below) including, plant derived by-products and microalgae as sustainable sources of alternative protein and bioactives such as, polyphenols, lipids and pigments. In particular, the selectivity of the separation, the underlying mechanism, and the impact of NADES on bioactivity and/or bioavailability are discussed.

### Extraction of phytochemicals

*Panic et al. (2021)* observed that polyol-based DES are excellent solvents for extraction of polyphenols considering their lightly acidic pH value (which is considered the optimum pH for polyphenols extraction) and low viscosity. Small differences in extraction efficiency among NADES tested were not surprising since they possessed similar physiochemical properties vital for extraction efficiency (see above).

On the other hand, *Panic et al. (2021)* discovered that all the examined NADES prove to be less effective as solvents for extracting D-limonene. It was surprising that the hydrophilic and polar solvent consisting of ChCl and glycerol with a 30% water content, managed to extract the lipophilic D-limonene compound at a concentration equivalent to approximately 60% of the concentration achieved with the hydrophobic reference solvent. They suggested that this phenomenon could be explained by the existence of microheterogeneity within the molecules comprising NADES. Additionally,

*Panic et al. (2021)* observed that in NADESs extracts, all other compounds found in orange essential oil (monoterpene and sesquiterpene hydrocarbons, esters, and aldehydes) were present in concentrations below detectable limits, in contrast to the results obtained from the hexane extract. Therefore, NADES-assisted extraction could be a promising approach for obtaining pure D-limonene.

In summary, research by *Panic et al. (2021)* indicated that polyol-based DES are effective solvents for polyphenol extraction, while NADESs have varying efficiency in extracting D-limonene. The surprising results regarding D-limonene extraction efficiency in some NADES may be related to the microheterogeneity of these solvents. This is turn agrees with *Choi et al. (2011)* who described NADES as a liquid capable of solubilizing intracellular compounds of intermediate polarity that neither dissolve in the lipid nor the water phase. *Ozturk, Parkinson & Gonzalez-Miquel (2018)* also found that the polyphenols from orange peels residues were efficiently extracted by ChCl NADES combined with glycerol or ethylene glycol and as efficiently or more than with ethanol. NADES enhanced extraction was ascribed to the like dissolves like principle as both, the extracted phenolic compounds and the selected NADES were considered polar. Moreover, *Ozturk, Parkinson & Gonzalez-Miquel (2018)* found that increasing water content reduced the extraction efficiency of NADES.

*Grillo et al. (2020)* found that the acidity of lactic, malic, and citric-based NADES proved advantageous for anthocyanin extraction and stability to storage, whereas glycerol-based NADES with a neutral or basic pH was unsuitable. ChCl:malic acid and ChCl:lactic acid led to the highest total anthocyanins yield, 23.41 and 23.59 mg/g, respectively. The evaluation of solvent-dependent degradation over nine months storage based on percentage polymeric colour (PPC) determination resulted in ChCl combinations with organic acids providing similar anthocyanin stability as the hydroalcoholic reference. In addition, *Grillo et al. (2020)* found that ChCl:lactic acid applied after microwave or ultrasound pretreatment significantly enhanced the anthocyanin extraction yield up to 25.83 and 21.18 mg/g of total anthocyanins within 15 and 30 min, respectively.

*Silva et al. (2020)* tested NADES extraction of anthocyanins from blueberry and compared with acidified methanol and aqueous extractions. Highest yield of anthocyanins was obtained with ChCl based NADES combined with citric acid although very similar yields were obtained with acidified methanol and water. However, NADES was considered advantageous in comparison to those solvents based on both, cost and reduced environmental impact. Furthermore, in a study by *Dai et al. (2016)* NADES was found to have a stabilization effect on the extracted anthocyanins. The degradation of one of the main anthocyanins, cyanidin, at different temperature and storage time was followed by liquid chromatography coupled with mass spectrometry and it was found that cyanidin had much higher stability in NADES (lactic acid:glucose) than in acidified ethanol particularly, when stored at −20 °C and to a lower extent at 4 °C. This stabilization effect was suggested to be due to the intermolecular interactions between the NADES components and the cyanidin resulting in its reduced oxidative degradation.

Moreover, NADES have been found to enhance the bioactivity and bioavailability of the extracts. *Wojeicchowski et al. (2021)* extracted phytochemicals from rosemary with NADES

based on ChCl and propylene glycol and they found that propylene glycol contributed to the antimicrobial activity of the extracts against both, Gram-positive and Gram-negative bacteria. Similarly, *Garcia-Roldan, Piriou & Jauregi (2022)* found that polyphenol extracted with NADES from a coffee by-product had much higher antimicrobial activity than the hydroethanolic and aqueous extracts; further evaluation of the antimicrobial activity of the NADES alone revealed that NADES contributed importantly to the measured activity and in some cases, synergistically. *Radošević et al. (2016)* obtained phenolic extracts of grape skin using ChCl based NADES and found that the ChCl:malic acid extract had an enhanced biological activity in comparison to the other NADES, based on antiproliferative tests against cancer cells; they suggested that this activity was due to the NADES forming compounds such as, malic acid. Furthermore, several authors have found that NADES have a beneficial effect on bioavailability of the extracted plant based bioactives (*Silva et al., 2020*; *Tong et al., 2021*) and on their bioaccesibility (*Zannou et al., 2022*).

### Extraction of proteins

Although little has been reported on the extraction of proteins NADES show promise as an extractant for proteins (see a recent review by *Zhou et al. (2022)*). One of the main attractive features of NADES is their fine-tuneability for specific purposes. As described above, changing the NADES components and their molar ratio as well as water composition results in changes in their physicochemical characteristics such as, pH. It is well known that pH is a very important parameter in the solubilization of proteins and will affect interactions between NADES components and the protein. In a study on the extraction of polyphenols from spent coffee ground by *Garcia-Roldan, Piriou & Jauregi (2022)*, it was found that ChCl:1,2-propanediol was more selective towards proteins than betaine:triethylenglycol and ethanol. It was hypothesized that protein extraction was favored due to the charge effect as choline is a positively charged quaternary amine. The effect of pH was further confirmed by *Panic et al. (2021)* who found that the pH value of the NADES tested affected to a great extent protein extraction ability. They found that the two NADES with the highest potential for protein extraction from orange peel were, glucose:glycerol and glucose:ethyleneglycol which, were nearly pH neutral (pH >6), while pH value of other tested NADES were mostly below 5.5. Moreover, they suggested that NADES could also enhance the extraction of proteins *via* the membrane disintegration effect either by membrane disruption and solvation of cytoplasmic proteins or by solvation of membrane integrated proteins that would not normally be soluble.

*Bai, Wei & Ren (2017)* also found that pH had a strong effect on protein extraction efficiency when applied ChCl based DES with different HBD compounds to the extraction of collagen peptides from cod skin instead of the conventional alkali/acid extraction. They found that the protein yield increased with the acidity of the HBD compound. In the cases of similar acidity, DES viscosity was the main factor influencing the protein yield as increased yield was found at lower viscosity. Their hypothesized that the ammonium salt formed between the hydrogen ions in DES (HBD) and imino group in proline and hydroxyproline aminoacids in collagen was the driving force for extracting collagen peptides from cod skin.

*Hernández-Corroto et al. (2020)* studied the extraction of protein from pomegranate peel by several ChCl-based NADES and compared against the extraction with pressurized liquid extraction (PLE) using ethanol as extractant. They found that at optimal conditions NADES protein extraction was superior (20 mg/g pomegranate peel) to PLE (9 mg/g pomegranate peel). They carried out further hydrolysis of the protein extracted by both methods and found that a higher number of peptides were identified in hydrolysates obtained from the NADES extract while a highernumber of phenolic compounds were observed in the hydrolysates obtained from the PLE extract than from the DES. So, in agreement with *Garcia-Roldan, Piriou & Jauregi (2022)* NADES demonstrated higher selectivity for proteins than ethanol whilst the latter favored the extraction of phenolic compounds.

*Xu et al. (2015)* investigated the use of DES as part of an aqueous two-phase system where, the top phase was a DES rich phase and the bottom phase a salt rich phase. ChCl-alcohol DES were used to investigate the partitioning of bovine serum albumin and trypsin in model solutions into the DES rich phase. They found that the extraction of the protein was driven mainly by the formation of aggregates and that electrostatic interactions were not the predominant driving forces. Moreover, they found that the conformation of the protein was not changed after extracted into DES-rich phase.

*Lin et al. (2020)* extracted protein from bamboo shoot, both from the edible part of the tip of the bamboo shoot (TBS), as well as its processing byproducts (basal bamboo shoot and sheath) using a DES comprised of compounds accepted as food additives: ChCl as HBA and levulinic acid as HBD. They found the yield of protein extraction from TBS was much higher with DES than with the conventional NaOH extraction. In addition, protein yield increased with DES components' molar ratio and water composition. Further increased in protein yield was obtained by repeating twice more the extraction which led them to suggest that extraction assisted technologies such as microwave or ultrasound could enhance extraction.

Often the back extraction of proteins can be problematic. As summarized by *Zhou et al. (2022)* proteins can be back-extracted by changing ionic strength, salt concentration and pH or adding ethanol to precipitate protein (Table 2). However, these methods have not been optimized and the back-extraction efficiency is typically very low.

There is still a lack of understanding of the mechanism of protein extraction and of the main interactions between NADES components and proteins driving this extraction.

### Microalgae

Microalgae are recognized as a promising source of bioactive compounds in food including, protein and their bioactive peptides, polyunsaturated fatty acids (EPA and DHA) and pigments (*Chen et al., 2022*). In recent years the interest in the exploitation of microalgae has rapidly grown due to their versatility and the ability to produce value-added compounds with applications in different sectors including, food, pharma, cosmetics, feed and fertilizers (*Mehariya et al., 2021*). In particular, the production of microalgae as an alternative protein source combined with the production of other products such as fatty acids, pigments and/or bioactive compounds following a biorefinery approach has attracted much attention among

**Table 2  Summary of the main methods used in NADES recovery and recycling.**

| NADES and application | Recovery method | NADES recovery and/or recycling | Reference |
|---|---|---|---|
| ChCl:citric acid | 1.NADES 25% H2O adsorption onto a resin | 1. NADES recovery 94,78%. | *Panic et al. (2019)* |
| Polyphenols from grape-pomace | 2. Prior dilution of NADES to 80% of water followed by adsorption | 2. NADES recovery 96,8%. | |
| ChCl:glycerol | Vacuum evaporation | NADES recycled once, twice and three times: 92%, 87%, and 81%, rutin recovery respectively. | *Huang et al. (2017)* |
| Rutin from tartary buckwheat hull | | | |
| ChCl:triethylene glycol | Solid–liquid extraction using C18 adsorbent | Data not available | Mansur et al. (2019) |
| Flavonoids from buckwheat sprouts | | | |
| Chcl:lactic acid | Vacuum evaporation | >96% of the NADES reagent was recovered. | *Kumar et al. (2018)* |
| Lignin and xylan from rice straw | | The solvent was reused in 3 cycles of biomass pretreatment and no significant difference in recovery of NADES were found | |
| ChCl:lactic acid | Adsorption | Recovery rate 93.98%. | *Xia, Li & Jiang (2021)* |
| Flavonoids from rhizomes of *Polygonatum odoratum* | | | |
| ChCl: 1,4-butanodiol | Adsorption follow by vacuum evaporation | Recovery yields of DES: 76.38%–91.15% | *Cui et al. (2018)* |
| Flavonoids from sea buckthorn leaves | | | |
| ChCl:glycerol | Back extraction of proteins: | Protein yield: 32.96% | *Xu et al. (2015)* |
| Bovine serum albumina | Precipitation by salting-out | No data available on NADES recovery | |
| ChCl:PEG 200 (ethylene glycol) | Back extraction of proteins: | Protein yield: 92.26%–97.97%. | *Liu et al. (2016)* |
| Pumpkin seed protein | Ethanol precipitation. | No data available on NADES recovery | |

the scientific and industrial community. One of the technological challenges is that these bioactives are typically produced intracellularly and therefore, their extraction will involve cell disruption as well as the use of non-green solvents such as, acetone for the extraction of pigments. Thus, the use of ecofriendly solvents such as NADES is a very attractive alternative. Only a few works have been reported on the use of NADES to extract a range of bioactives from microalgae as reviewed by *Mehariya et al. (2021)*. In this review, the importance of applying a pretreatment (*e.g.*: microwave, high pressure homogenization, sonication, enzymatic hydrolysis) prior to NADES extraction is highlighted to partially disrupt the cell wall and hence, facilitate bioactives extraction. Furthermore, *Moldes et al. (2022)* highlight in their review the few works that have been published on the application of DES to protein recovery.

*Cicci, Sed & Bravi (2017)* reported the successful extraction of chlorophylls and carotenoids from microalgae by 1,2-propanediol:ChCl:water (1:1:1 molar ratio) but less successful extraction of nonpolar compounds such as lipids. In addition, in the studies by *Hilali et al. (2024)* and *Wils et al. (2021)*, it was demonstrated that hydrophilic NADES, specifically the glycerol:glucose (2:1) NADES with 20% water, had the ability to extract a combination of valuable compounds including, antioxidant phycocyanin, carotenoids, and free fatty acids (FFA). *Wils et al. (2021)* showed that the addition of water (10% and 20%) to that system significantly increased the extraction of phycocyanin, chlorophyll, carotenoids, and FFA. They suggested that this enhanced extraction might be attributed to improved mass transfer due to lower viscosity or better access to cellular content *via* hydration of the biomass. Moreover, the study conducted by *Hilali et al. (2024)* found that glycerol:glucose NADES, a sugar based NADES which is less polar than water and with polarity close to methanol, could extract a significant quantity of FFA. The authors suggested that this unexpected capability might be attributed to the solvent's duality, wherein it possesses both lipophilic and hydrophilic properties. This duality of the solvent is also supported by the research of *Choi et al. (2011)* and *Panic et al. (2021)*, as described above (see "Extraction of phytochemicals" above) .

*Wils et al. (2021)* found that some hydrophobic NADES, particularly those based on menthol or FFA, demonstrated an increased extraction capacity for FFA when compared to the reference solvents. Additionally, *Hilali et al. (2024)* observed that with menthol-based NADES, even though the overall recovery was lower compared to FFA-based NADES, improved selectivity towards polyunsaturated fatty acids (PUFA) was obtained. Moreover, *Hilali et al. (2024)* found that menthol based NADES (menthol:1,2-Octanediol) had the highest selectivity for non-polar pigments.

On the other hand, *Lu et al. (2016)* investigated the application of aqueous DES as a pretreatment for lipid extraction from *Chlorella sp.* They used a mixture of ethyl acetate and ethanol as a solvent and found that pretreatment with aqueous DES (aqueous ChCl:oxalic acid, aqueous ChCl:ethylene glycol and aqueous urea:acetamide) led to significant improvements in lipid recovery rates compared to untreated biomass. The improvement in lipid recovery was attributed to the enhancement of the permeability of the algal cell wall. The authors proposed that this enhancement was a result of the reduction in the inner molecule hydrogen bond energy of macromolecules present in the cell wall, such as cellulose and hemicellulose. The aqueous DES formed hydrogen bonds with these macromolecules in the algal cell wall, altering their hydrogen bond formation and making the cell wall more permeable.

Clearly NADES have shown promise in the extraction of a wide range of valuable compounds from microalgae, including lipids, pigments, and fatty acids. The addition of water to certain NADES systems, such as glycerol:glucose, has been shown to enhance the extraction of these compounds, possibly due to improved mass transfer and cell wall disruption. The duality of some NADES, exhibiting both lipophilic and hydrophilic properties, makes them versatile solvents for these applications. Furthermore, the use of aqueous DES as a pretreatment method for microalgae has demonstrated significant improvements in lipid recovery rates. In conclusion, these studies collectively highlight

the potential for developing efficient and sustainable processes for extracting bioactive compounds from microalgae.

### Recyclability of NADES

While NADES offer several advantages, such as low toxicity, biodegradability, and being composed of cheap and abundant components, they do have limitations and challenges regarding their recyclability:

#### Separation efficiency

Some NADES may have difficulty in separating from the solutes they dissolve, making it challenging to recover the solvent in its pure form after use. This can decrease the efficiency of recycling methods. *Karadendrou et al. (2022)* reported that proline:glycerol 1:2 NADES was recovered after vacuum evaporation of the filtrate and was then reused without any further purification for up to six times without any significant loss in yield. In another study (*Wang et al., 2021*), the acidic NADES (ChCl:oxalic acid) used for the extraction of resveratrol from *Polygonum cuspidatum* was successfully recycled three time after simple extraction with ethyl acetate.

#### Thermal stability

Many NADES may not be stable at high temperatures, limiting their application in processes that require elevated temperatures. This instability can also affect their recyclability as they may degrade during recycling processes involving heating.

#### Energy intensive processes

Some recycling methods for NADES may require energy-intensive processes such as distillation or extraction, which could counteract the environmental benefits of using renewable solvents.

#### Purity concerns

Contaminants from the solutes dissolved in NADES may affect the purity of the recovered solvent, requiring additional purification steps that could increase the cost and complexity of recycling.

Despite these limitations, researchers are actively working on developing methods to improve the recyclability of NADES. Some strategies being explored include techniques such as membrane filtration, adsorption, or precipitation, which could enable the efficient separation and recovery of NADES from solutes (see summary of main strategies in Table 2). Alternatively, NADES can be considered a formulating aid given its stabilization effect and/or contribution to bioactivity however, the cost effectiveness of this approach will need to be assessed based on the added value of the extracted compounds.

## CONCLUSIONS

One of the main attractive features of NADES is their fine-tuneability for specific purposes. Extraction efficiency can be maximised by adjusting the physicochemical properties of NADES (polarity, pH, viscosity) to those that favour the solubility of the target molecule

following the principle of like dissolves like but also by favouring interactions between the target molecule and NADES components. In addition, NADES unlike the organic solvents have the unique ability to partially disrupt plant and microalgae cell walls which results in increased permeability and improved extraction efficiency. Moreover, NADES have proven to have a stabilising effect on anthocyanins and can enhance biological activity and bioavailability of the extracted bioactives. These properties together with their low environmental impact particularly, in terms of their low toxicity and high biodegradability makes NADES very attractive for biorefinery applications.

Although NADES have shown promise on the extraction of bioproducts and there is an improved understanding of the mechanism of extraction yet, there is a need for improved insight into the selectivity and in particular, for an improved understanding of protein extraction and back extraction. Moreover, the recyclability of NADES and environmental impact assessment of the extraction processes are important aspects to consider for assessing scalability and potential industrial applications. Further advances on modelling for predicting extraction efficiency and/or interactions with a target molecule will also contribute to take full advantage of this extraction method for a wide range of bioproducts in biorefinery processes.

### Funding
The Basque Government—Department of Economic Development, Sustainability and Environment—Vice. Dept. of Agriculture, Fishing and Food Policy, Directorate of Quality and Food Industries provided funding for the project and the scholarship of Leire Esnal-Yeregi. The funders had no role in study design, data collection and analysis, decision to publish, or preparation of the manuscript.

### Grant Disclosures
The following grant information was disclosed by the authors:
Basque Government—Department of Economic Development, Sustainability and Environment—Vice. Dept. of Agriculture, Fishing and Food Policy, Directorate of Quality and Food Industries.

### Competing Interests
The authors declare there are no competing interests.

### Author Contributions
- Paula Jauregi conceived and designed the experiments, performed the experiments, analyzed the data, prepared figures and/or tables, authored or reviewed drafts of the article, and approved the final draft.
- Leire Esnal-Yeregi conceived and designed the experiments, performed the experiments, analyzed the data, prepared figures and/or tables, authored or reviewed drafts of the article, and approved the final draft.

- Jalel Labidi conceived and designed the experiments, authored or reviewed drafts of the article, and approved the final draft.

## Data Availability

This is a literature review.

## Supplemental Information

Supplemental information for this article can be found online at http://dx.doi.org/10.7717/peerj-achem.32#supplemental-information.

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
