# Peer review of "Natural deep eutectic solvents (NADES) for the extraction of bioactives: emerging opportunities in biorefinery applications"

_PeerJ Analytical Chemistry, doi:10.7717/peerj-achem.32_

## Round 0.1 · original submission · Major Revisions

Dear authors,
Reviewers have now commented on your submission, and you will see that they have raised major concerns. I am prepared to consider a revised version of the manuscript that addresses the raised concerns of the reviewers.

Since this is a review, your draft should explain explicitly what contributions this submission makes to the already known literature, how it builds on previous reviews in this area, or why the review is timely and required at this point in time.

After a careful and exhaustive review of published literature, the study should lead to meaningful conclusions and identify areas for future research that can guide the field further.

**Language Note:** The review process has identified that the English language must be improved. PeerJ can provide language editing services - please contact us at copyediting@peerj.com for pricing (be sure to provide your manuscript number and title). Alternatively, you should make your own arrangements to improve the language quality and provide details in your response letter. – PeerJ Staff

Reviewer 1 ·

Basic reporting

1) The document does not meet the requirements outlined in the PeerJ author guidelines, such as the line number, for instance.

2) English writing needs improvement, there are some spelling mistakes.

3) The references presented are from recent years, however, they seem insufficient, taking into account the growing number of research projects in the field of the use of NADES as extraction solvents. Additionally, the point of view from which the topic presented in the document is analyzed does not offer new information to what has already been analyzed in several review articles.
Here are some examples:
https://doi.org/10.3390/plants10102091
https://doi.org/10.4028/www.scientific.net/KEM.797.20
https://doi.org/10.3390/ijms23063381
https://doi.org/10.3390/ijms23063381
https://doi.org/10.1016/j.aca.2018.07.059
https://doi.org/10.1021/acs.jafc.0c06641
https://doi.org/10.3390/app9194169
https://doi.org/10.3390/app12052391
https://doi.org/10.1016/j.trac.2023.117410
https://doi.org/10.1080/10408347.2021.1946659

4) On the other hand, in this review, NADES are presented as extraction solvents that supposedly offer environmental advantages over conventional solvents, but the limitations of their use are not analyzed, such as the recovery and purification of the desired molecules.

5) The introduction lacks clarity regarding its intended audience and the necessity of this document. Furthermore, it doesn't effectively explain the rationale behind selecting polyphenols, proteins, and compounds derived from microalgae as subjects of study.

Experimental design

1) The Survey Methodology presented seems incomplete, it does not describe in detail how the presented topic was selected, and the period that was considered to cover the research related to the selected topic. Nor is it presented how the topics presented were organized, nor the process followed for the review of the topic and articles selected for the writing of this document.

2) All incorporated information was cited and referenced throughout the manuscript, maintaining a uniform citation style.

3) The manuscript is organized into coherent sections, however, the information provided is brief, for example in the section on the physicochemical properties of NADES it is not clear because only the viscosity, polarity, pH, and melting point are included, but not the density or conductivity, for example.

Validity of the findings

1) The relevance of this manuscript is not justified since it is limited to describing some research related to the selected topic; however, the gap that is intended to be covered with this review is not clearly defined.

2) The conclusion presented does not seem to provide a conclusive analysis of the state of the art by the authors.

Additional comments

I believe that this manuscript requires careful review before being considered for publication. It should provide a broader perspective on the subject matter and conduct an in-depth analysis of the advantages and disadvantages of using NADES as solvents. Currently, the manuscript does not offer any significant or unique insights into the chosen topic, in comparison to other existing literature.

Annotated reviews are not available for download in order to protect the identity of reviewers who chose to remain anonymous.

Reviewer 2 ·

Basic reporting

1. The current abstract is quite detailed and dense. I suggest streamlining it to focus on the key findings, avoiding overly technical details better suited for the main text. Organizing the abstract into distinct, logical segments and employing simpler language will significantly improve readability.

2. The introduction would benefit from a broader context regarding the significance of NADES in environmental and industrial scenarios. A brief discussion on the limitations of traditional solvents and the urgency for greener alternatives would set a strong foundation for your review.

3. When introducing specific terms like 'circular approaches' and 'biorefinery concept,' including references at their first mention will enhance clarity, especially for readers less familiar with these concepts.

4. The manuscript requires meticulous proofreading to correct grammatical errors and enhance language clarity. Some sections are currently challenging to comprehend, potentially hindering reader understanding. Assistance from a fluent English speaker in refining the manuscript could significantly improve readability.
Some typos on Page 6 are listed as follows:
Line 20: Remove the extraneous symbol after "components."
Line 24: Clarify "choline chloride o betaine."
Line 29: Add 'and' before "viscosity."

Experimental design

More detail on the selection criteria for your survey methodology would be beneficial. Clarifying how you determined the relevance and quality of the information presented will strengthen your research approach.

Validity of the findings

1. For the section on NADES in biorefinery applications, rather than summarizing key features in bullet points, a deeper exploration of these features would be more informative. Additionally, a detailed discussion on how NADES enhance the stability and bioactivity of bioactives would enrich this section.

2. A section discussing potential future directions for NADES research would be a valuable addition. Consider including emerging technologies, interdisciplinary approaches, and potential industrial applications.

Reviewer 3 ·

Basic reporting

no comment

Experimental design

no comment

Validity of the findings

no comment

Additional comments

The manuscript demonstrates a clear potential to contribute to the existing body of knowledge in the field. The manuscript is well organized and well written in general. The cited references are relevant and up to date. I recommend publication after correcting some minor typographical errors that some of them listed below as example;

L23 “….choline chloride o betaine with…” what “o” stands for?
L33 “…have a stabilizing effect…”
L80: choline chloride,…
L161: please use abbreviation
L164: Viscosity is not only related to water content

---

## Round 0.2 · Minor Revisions

Dear Authors,

The re-evaluation of your manuscript has identified some minor corrections which need to be made. Please carry out the suggested revisions.

Reviewer 1 ·

Basic reporting

English still can be improved.
There are some mistakes such as PH in line 280, it must be "pH".

Experimental design

No comment

Validity of the findings

No comment

Additional comments

A graphical abstract should be added.
Still it requires acronyms homogenization: it is DES or NADES

Reviewer 2 ·

Basic reporting

The manuscript has effectively addressed several concerns raised in the previous review cycle. However, there remain certain aspects requiring further elucidation:

The relationship between density, viscosity, and temperature appears somewhat unclear and disordered. It is recommended to either restructure the paragraph or incorporate a graphical representation to illustrate these relationships and associated physicochemical properties more clearly.

Experimental design

N/A

Validity of the findings

N/A

Additional comments

Additionally, there are a few typos that need correcting in the sentences:

- "Even ethanol, which is considered a green solvent and therefore can be used in food applications, has the disadvantage of high flammability and low selectivity of extraction because of its high solvating power." (Remove the unnecessary comma after "ethanol" and place the second comma after "therefore" in the sentence discussing ethanol's properties.)

- "Therefore, NADES are gaining much interest for biorefinery applications, including the recovery of bioactives." (Add a comma after "including" and pluralize "bioactive" to "bioactives".)

- "Much attention has been given to this topic in recent years; however, there is not much published yet on the extraction of proteins and the extraction of bioactives from microalgae, which is a promising and sustainable source of bioactives." (Insert a comma after "years," and replace "which, is" with ", which is".)

Addressing these issues (some are not listed above) will further enhance the clarity and professionalism of the manuscript.

---

## Round 0.3 · accepted · Accept

Dear Authors,

Thank you for revising your manuscript as requested by the reviewers.
It is now in a form that will be of benefit to the scientific community and provide areas for future research